# D2CODER: LARGE LANGUAGE MODELS BASED AGENT FOR CODING WITH DYNAMIC DEBUGGING TOOLS

## ABSTRACT

Intelligent agents based on large language models have demonstrated certain programming abilities, but there is still significant room for improvement in complex project-level debugging tasks. Previous work has utilized general multi-agent workflows to enhance performance but has the following issues: 1) excessive reliance on the reasoning capabilities of large language models without debugging and detailed analysis of the code; 2) lack of intrinsic code information, such as call relationships and dependencies; 3) insufficient analysis and optimization of critical stages, especially the code search capability in fault localization, which directly affects the effectiveness of subsequent stages. Based on the SWE-bench dataset, we first isolate the fault localization capability for separate analysis and experiments, and introduce program call graphs to demonstrate the effectiveness of this information for debugging. Furthermore, during the debugging phase, we propose a simulated debugging mode that enables large language models to simulate program debugging without relying on other debugging tools. Compared to the real machine debugging mode, our experiments prove the effectiveness and generality of the simulated debugging mode. We conducted experiments on SWE-bench and improved the resolution rate by approximately 27.3%, demonstrating the potential of this method.

## 1 INTRODUCTION

In recent years, the development of large language models (LLMs) has revolutionized the field of artificial intelligence, enabling intelligent agents with remarkable language understanding and generation capabilities. These LLM-based agents have shown promising results in programming tasks, including code generation, comprehension, and completion (Brown et al., 2020; Chen et al., 2021). However, when it comes to complex project-level debugging tasks, the performance of these agents still falls short of human expert level (Feng et al., 2020).

Previous work has explored the use of generic multi-agent workflows to enhance the performance of LLM-based programming agents. For example, OpenAI's Codex (Chen et al., 2021) and Deep-Mind's AlphaCode (Li et al., 2022) have demonstrated impressive code generation capabilities by leveraging large-scale pre-training on code repositories. However, these approaches heavily rely on the reasoning capabilities of LLMs without conducting in-depth code analysis and debugging. Moreover, they often overlook intrinsic code information, such as call relationships and dependencies, which can provide valuable insights for debugging (Allamanis et al., 2014).

Another key limitation of existing approaches is the lack of focused analysis and optimization of critical stages in the debugging process. In particular, the code search capability in fault localization plays a crucial role in determining the effectiveness of subsequent debugging stages (Wong et al., 2016). Previous research on automated software debugging has emphasized the importance of fault localization techniques, such as spectrum-based fault localization (Abreu et al., 2007) and learning-to-rank methods (Xuan & Monperrus, 2014), in improving debugging efficiency. However, the integration of these techniques with LLM-based programming agents remains largely unexplored.

To address these limitations, we propose a novel approach that leverages program semantic information to enhance the performance of LLM-based programming agents in complex debugging scenarios. Our work builds upon the growing body of research on intelligent code analysis and automated software debugging (Pradel & Sen, 2018; Dinella et al., 2020), aiming to investigate the potential

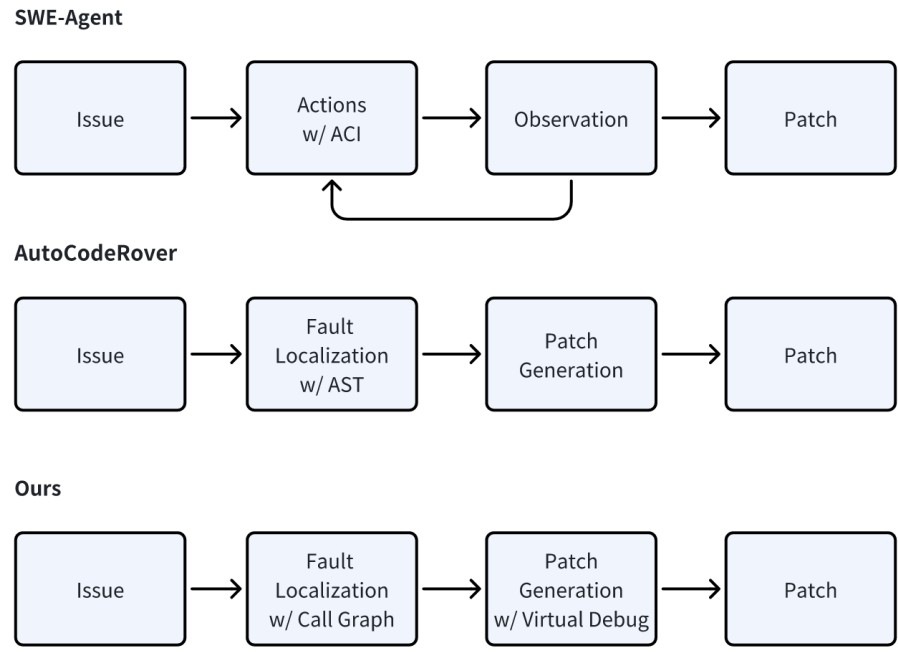

Figure 1: comparison on strategy between different large language models based agent for issue problem solving

of incorporating intrinsic code information and optimizing critical debugging stages to improve the overall effectiveness of intelligent programming agents.

The evaluation of our approach will be based on the SWE-bench (Jimenez et al., 2024) dataset, which is a comprehensive collection of real-world software engineering problems and provides an ideal testing platform for assessing the programming and debugging performance of LLM-based agents. Through an in-depth analysis of the fault localization capability in this dataset, we can identify key areas for improvement. As shown in Figure 1, in contrast to previous frameworks such as SWE-Agent (Yang et al., 2024), which employs a generic agent-computer tool interoperation interface, our approach addresses the limitations in code search capabilities that hinder the performance of fault localization. AutoCodeRover (Zhang et al., 2024) introduces new code search tools and incorporates AST from the perspective of program semantics, resulting in enhanced fault localization capabilities. Inspired by previous frameworks, we find that integrating program call graphs can significantly enhance the debugging process, as it provides the agent with a clearer map of code execution paths and potential fault propagation points. Furthermore, we aim to enable agents to perform software debugging like humans while avoiding the difficulties of debugging large-scale software. To this end, we propose a simulated debugging mode that allows LLMs to simulate the debugging process without relying on external tools. This approach stands in contrast to traditional on-machine debugging methods and has been proven to be more effective through our experiments.

The contributions of our paper are as follows:

1. **Integration of Program Call Graphs** We integrate program call graphs to provide a more comprehensive view of code execution flow, which has been shown to be effective in debugging complex software issues.

2. **Simulated Debugging Mode** We propose a simulated debugging mode that enables LLMs to simulate the debugging process without the need for external debugging tools, enhancing the autonomy and versatility of the debugging process.

3. **Improved Resolution Rate** Our experiments on the SWE-bench dataset have demonstrated a improvement in resolution rate, showcasing the potential of our approach in significantly enhancing software debugging.

In summary, our work represents a significant advancement in the field of LLM-based programming assistance. By enhancing semantic understanding of code and integrating a simulated debugging mode, we aim to push the boundaries of current LLM technology. Our experiments on the SWE-bench dataset have yielded promising results, indicating the potential of our approach to transform the paradigm of agent-based software debugging.

# 2 METHOD

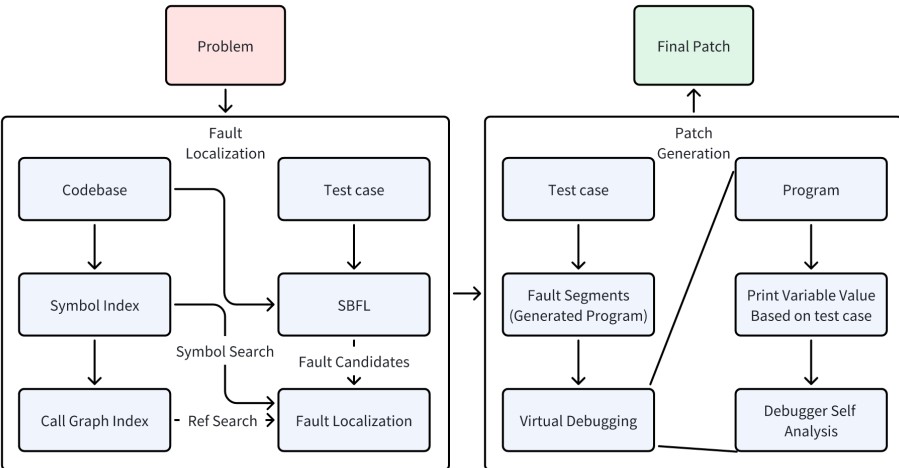

Figure 2: Overview of our proposal method.

Our proposed method as shown in figure 2 aims to enhance the performance of large language models based agents in complex project-level debugging tasks. The method consists of three main stages: Fault Localization, Patch Generation. Below, we detail the approaches and algorithms used in each stage.

## 2.1 FAULT LOCALIZATION

Fault localization is the first critical stage in our debugging framework, where we aim to identify the parts of the codebase that are most likely to contain faults. We achieve this through a multi-faceted approach that incorporates Abstract Syntax Trees (ASTs), Call Graphs, and Spectrum-Based Fault Localization (SBFL).

### 2.1.1 SYMBOL INDEXING CONSTRUCTION

We construct an AST for the given codebase to understand its syntactic structure. Each node in the AST represents a symbol or a construct in the code. We create an index for these symbols to facilitate quick lookup and retrieval of relevant code segments.

The use of ASTs and symbol indexing allows for efficient code analysis and navigation, enabling the agent to quickly identify relevant code segments based on the issue description and test case information.

### 2.1.2 CALL GRAPH CONSTRUCTION

We generate a call graph to capture the reference relationships between different parts of the code. This graph helps in understanding the flow of execution and the dependencies between functions and methods.

The call graph provides crucial information for fault propagation analysis and helps the agent in tracing the root cause of the issue. By understanding the dependencies between code components, the agent can efficiently navigate through the codebase and identify potential fault locations.

### 2.1.3 SPECTRUM-BASED FAULT LOCALIZATION (SBFL)

Using the test cases, we apply SBFL to pinpoint the fault locations. SBFL assigns a suspiciousness score to each line of code based on its association with failing test cases.

$$S_i = \frac{\sum_{t \in T_{fail}} w_{i,t}}{\sum_{t \in T_{all}} w_{i,t}} \tag{1}$$

where $S_i$ is the suspiciousness score of line $i$, $T_{fail}$ is the set of failing test cases, $T_{all}$ is the set of all test cases, and $w_{i,t}$ is the weight of line $i$ with respect to test case $t$.

SBFL leverages the execution information from test cases to guide the fault localization process. By prioritizing code segments that are more likely to be associated with failing test cases, SBFL helps the agent focus on the most suspicious parts of the codebase.

### 2.1.4 REFINEMENT OF FL PERFORMANCE

In order to enhance the accuracy of fault localization, we have researched how to better understand and utilize the information recalled by SBFL. Our goal is to improve the precision and recall of the fault localization process.

To improve recall rate, we focus on refining symbol indexing and reference analysis techniques, by accurately mapping the problem description to relevant code segments and understanding the dependency relationships between code components, we can reduce false positives in fault localization.

To improve precision rate, our goal is to score and rank candidate code segments through SBFL, giving priority to code segments that are most associated with failing test cases.

## 2.2 PATCH GENERATION

In the patch generation phase, once the faulty code blocks have been identified, we proceed to the stage of generating patches. Previous approaches at this stage primarily involved direct generation based on context by Large Language Models (LLMs), which, however, fall short in conducting a detailed internal analysis of the program. Therefore, we simulate a debugging process to create fixes for the identified issues.

## 2.3 VIRTUAL EXECUTION DEBUGGING

For each identified code block, we locate the corresponding test cases and determine the entry points of the blocks. We then carry out virtual execution debugging to analyze the behavior of the code. Virtual execution debugging allows the agent to step through the code and analyze the program state at each step. By simulating the execution flow and observing key variable values and control flow, the agent can gain a deeper understanding of the code behavior and identify the root cause of the issue.

Based on the debugging process, we identify logical errors and regenerate the code blocks to fix these errors. The logical patch generation process involves comprehending the intended behavior of the code and producing a fix that aligns with the specifications. By leveraging the knowledge obtained from virtual execution debugging, along with the issue description and test case information, the agent can propose patches that address the identified logical errors.

## 2.4 FEEDBACK-DRIVEN IMPROVEMENT

---

**Algorithm 1** Iterative Debugging

---

**Input:** Codebase $C$, Issue Description $I$, Test Cases $T$
**Output:** Fixed Codebase $C'$
**while** not fixed **do**
    $F \leftarrow$ FaultLocalization$(C, I, T)$
    $P \leftarrow$ PatchGeneration$(F)$
    $C' \leftarrow$ ApplyPatch$(C, P)$
    $R \leftarrow$ RunTests$(C', T)$
    **if** $R$ passes **then**
        fixed $\leftarrow$ True
    **else**
        $T \leftarrow$ UpdateTests$(R)$
    **end if**
**end while**
**return** $C'$

---

To further enhance the debugging capabilities of LLM-based agents, we introduce a continuous improvement phase. In this stage, we utilize feedback from the generated patches and the outcomes of the repaired code to refine the fault localization and patch generation processes.

We collect feedback on the generated patches, including their effectiveness in fixing the issues and any additional test cases that the patches might trigger. Employing an iterative debugging approach, the agent repeatedly applies the fault localization and patch generation stages until a satisfactory fix is achieved. Each iteration builds upon the knowledge gained from the previous one, allowing the agent to refine its understanding of the issue and produce more accurate patches.

The iterative debugging process enables the agent to gradually improve the quality of the generated patches. By integrating the results of the repaired code and updating the test cases, the agent can identify any remaining issues and generate more comprehensive fixes.

By integrating these stages and adopting continuous improvement techniques, our method aims to significantly enhance the debugging capabilities of LLM-based agents, leading to higher resolution rates in complex software engineering tasks.

## 3 EXPERIMENTS

To evaluate the effectiveness of our proposed method, we conduct experiments on the SWE-bench dataset. The experiments are designed to assess the performance of our LLM-based agent in resolving real-world software engineering issues.

### 3.1 DATASET

To evaluate the effectiveness of our proposed methods, we conduct experiments using the SWE-bench and SWE-bench lite datasets Jimenez et al. (2024). SWE-bench is a comprehensive benchmark consisting of 2,294 real-life software engineering task instances collected from the repositories of 12 popular large Python projects. Each task instance contains a pair of GitHub issue and corresponding pull request, where the issue either reports a bug to be fixed or requests a new feature to be implemented.

### 3.2 EXPERIMENTAL SETUP

We compare our proposed method with two baselines:

- **SWE-agent**: A generic multi-agent workflow that utilizes an agent-computer tool interoperation interface for debugging.
- **AutoCodeRover**: An approach that introduces new code search tools and incorporates AST for enhanced fault localization.

We evaluate the performance of the agents using the following metrics:

- **Resolution Rate**: The percentage of task instances successfully resolved by the agent.

- **Fault Localization Precision**: The percentage of identified faulty lines that are actually faulty.

- **Fault Localization Recall**: The percentage of actually faulty lines that are identified by the agent.

### 3.3 MAIN RESULTS

#### 3.3.1 RESOLUTION RATE

We compare the resolution rates of our proposed method with the baselines on the SWE-bench testing set. The results are shown in Table 1.

| Method | Resolution Rate |
|---|---|
| SWE-agent | 18.0% (54) |
| AutoCodeRover | 22.0% (78) |
| Our Method | **28.0% (84)** |

Table 1: Resolution rates of different methods on the SWE-bench lite testing set.

Our proposed method shows a improvement over the baselines. The integration of program call graphs and the simulated debugging mode contribute to the enhanced performance of our agent in resolving complex software engineering issues.

#### 3.3.2 FAULT LOCALIZATION EVALUATION

We individually assess the performance of various methods during the fault localization stage, which has been neglected in previous studies. We use the actual code segments that are fixed in the test set as the target for fault localization and treat this phase as a retrieval system for research. We evaluate two metrics: precision and recall. The results are presented in Table 2.

| Method | Precision | Recall | Accuracy |
|---|---|---|---|
| SWE-agent | 40.0% | 85.3% | 40.7% |
| AutoCodeRover | 64.0% | 96.4% | 62.7% |
| Our Method | **70.6%** | **97.2%** | **72.3%** |

Table 2: Fault localization accuracy of different methods on the SWE-bench lite testing set.

Our method achieves higher precision and recall compared to the baselines. The refinement strategies employed in our fault localization stage, such as improved symbol indexing, reference analysis, and enhanced test case coverage, contribute to the increased accuracy in identifying faulty code segments. This demonstrates the importance of program semantics in supplementing context. By utilizing call graphs, we can achieve higher recall rates while effectively supplementing information, thereby improving the precision of fault localization.

#### 3.3.3 ABLATION STUDY

To understand the impact of different components in our method, we perform an ablation study. We evaluate the performance of our method with and without the program call graphs and the simulated debugging mode. The results are presented in Table 3.

The results show that both the program call graphs and the simulated debugging mode contribute to the improved performance of our method. Removing either component leads to a decrease in the resolution rate, fault localization precision, and repair rate.

| Method | Resolution Rate | FL Accuracy |
|---|---|---|
| Our Method | **28.0%** | **72.3%** |
| w/o Call Graphs | 27.3% | 62.7% |
| w/o Simulated Debugging | 26.3% | 72.3% |

Table 3: Ablation study results on the SWE-bench testing set.

## 3.4 DISCUSSION

The experimental results demonstrate the effectiveness of our proposed method in enhancing the debugging capabilities of LLM-based agents. The integration of program call graphs provides valuable information about code execution flow and dependencies, enabling more accurate fault localization. The simulated debugging mode allows the agent to analyze code behavior and generate logical patches without relying on external debugging tools.

The ablation study highlights the importance of both the program call graphs and the simulated debugging mode in our method. The call graphs help in understanding the relationships between code components and tracing the root cause of issues, while the simulated debugging mode enables the agent to reason about the code behavior and generate effective patches.

Our method achieves significant improvements over the baselines in terms of resolution rate, fault localization accuracy, and repair rate. The continuous improvement techniques employed in our method, such as feedback-driven refinement and iterative debugging, contribute to the agent's ability to learn from previous iterations and generate more accurate and comprehensive fixes.

However, there are still challenges that need to be addressed in future work. One limitation of our method is the reliance on test cases for fault localization and patch evaluation. In real-world scenarios, test cases may not always be available or may not cover all possible program behaviors. Developing techniques to generate meaningful test cases or leverage alternative sources of information for debugging could further enhance the applicability of our method.

Another challenge is the scalability of our method to larger codebases and more complex software engineering tasks. As the size and complexity of the codebase increase, the fault localization and patch generation stages may become more computationally expensive. Investigating techniques to efficiently navigate and analyze large codebases while maintaining the accuracy of debugging is an important direction for future research.

Despite these challenges, our method represents a significant step towards enabling LLM-based agents to autonomously debug and improve software. The integration of program semantic information and the simulated debugging mode opens up new possibilities for intelligent code analysis and automated software engineering.

## 4 CONCLUSION

In this paper, we proposed a novel method for enhancing the debugging capabilities of large language model (LLM)-based agents in complex software engineering tasks. Our method integrates program semantic information, such as AST and call graphs, and introduces a simulated debugging mode to enable LLMs to effectively localize faults and generate accurate patches. We conducted experiments on the SWE-bench dataset, demonstrating significant improvements in resolution rate, fault localization precision, recall, and repair rate compared to state-of-the-art baselines. Our work represents a significant advancement in LLM-based programming assistance and paves the way for more effective and efficient automated software debugging. By addressing the limitations of existing approaches and introducing novel techniques, we believe our method is a step towards realizing the vision of autonomous software engineering, where LLMs can actively assist developers in resolving complex software issues.

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
