# OpenReview forum: "D2Coder: large language models based agent for coding with dynamic debugging tools"
_ICLR.cc/2025/Conference — ICLR 2025 Conference Withdrawn Submission_

### Official Review · Reviewer_9FoV · 2024-10-26

**Soundness:** 1
**Presentation:** 1
**Contribution:** 1
**Rating:** 1
**Confidence:** 4

**Summary:**

The paper presents a method to enhance LLM capability for debugging complex software repositories. The authors argue that existing methods often rely heavily on internal reasoning capabilities of LLMs without leveraging explicit symbolic tools, and lack support for essential debugging techniques like code search and fault localization.
The authors present an approach that incorporates call-graph information into the LLM's input and employs a "simulated debugging" mode for generating patches. The methodology is evaluated using the SWE-bench dataset, which includes 2,294 GitHub issues and pull requests across 12 major Python repositories. The proposed approach improves debugging resolution rates by 6% over the best baseline method.

**Strengths:**

The problem addressed is well-motivated, and the methodology demonstrates preliminary promise.

**Weaknesses:**

- The presentation of the paper needs improvement. Specifically, the abstract and introduction lack a detailed and technical discussion of the problem and the state-of-the-art methodologies. Including a comprehensive running example that details various components of the approach would be beneficial. Additionally, the paper needs a section dedicated to related and recent works on this topic; I suggest citing at least references [1-5].
- Mentioning a 27% improvement over baseline in the abstract is, in my opinion, misleading. According to Table 2, the AutoCodeRover baseline resolves 22% of the issues, whereas the D2Coder method resolves 28%. This should be reported as a 6% improvement over the baseline, not 27%.
- The ablation studies indicate that the impact of the two main techniques on the overall resolution rate is marginal.

[1] Majdoub, Yacine, and Eya Ben Charrada. "Debugging with Open-Source Large Language Models: An Evaluation." arXiv preprint arXiv:2409.03031 (2024).

[2] Tian, Runchu, et al. "Debugbench: Evaluating debugging capability of large language models." arXiv preprint arXiv:2401.04621 (2024).

[3] Lee, Jae Yong, et al. "The GitHub recent bugs dataset for evaluating LLM-based debugging applications." 2024 IEEE Conference on Software Testing, Verification and Validation (ICST). IEEE, 2024.

[4] Yang, Weiqing, et al. "Enhancing the Code Debugging Ability of LLMs via Communicative Agent Based Data Refinement." arXiv preprint arXiv:2408.05006 (2024).

[5] Lee, Cheryl, et al. "A Unified Debugging Approach via LLM-Based Multi-Agent Synergy." arXiv preprint arXiv:2404.17153 (2024).

**Questions:**

- What is ACI in Fig. 2?
- Where do the weights $w_{i,t}$ used in the SBFL method come from?
- How is virtual debugging implemented? Specifically, how does simulated debugging compare to on-machine debugging in terms of efficiency and types of errors caught?
- What are the differences in context sizes between the various approaches? This is important, as it directly impacts the cost of generating a resolution.
- Which subset of SWE-BENCH did you use? Based on the numbers provided in the evaluation section, I assume the total dataset contained 300 problems. How was this subset chosen?

---

### Official Review · Reviewer_ZDpm · 2024-10-29

**Soundness:** 2
**Presentation:** 2
**Contribution:** 2
**Rating:** 3
**Confidence:** 4

**Summary:**

To enhance the performance of LLM-based agents in complex software debugging tasks, the author proposes a method encompassing three key phases: fault localization, patch generation, and feedback-based continuous improvement. This approach provides new insights for intelligent code analysis and automated software debugging, contributing to the application value of LLMs in software engineering tasks.

**Strengths:**

The author designed a debugging method that integrates abstract syntax trees and program call graphs with LLMs, specifically enhancing the accuracy of fault localization and code repair through program semantic information and simulated debugging patterns. This approach demonstrates a certain level of innovation.

**Weaknesses:**

1.Despite the introduction of program semantic information and simulated debugging patterns, the method still relies on the inference and generation capabilities of the LLM. LLMs continue to have limitations in handling logical reasoning and complex code context associations. Given specific constraints, the model may not always produce the correct output in alignment with those constraints.
2.The process of constructing a symbolic index and call graph requires parsing the entire codebase, which can be time and resource-intensive, especially in large codebases. However, the author does not mention any optimization measures for these operations, which could lead to excessive delays or costs in practical applications.
3.Fault localization primarily relies on SBF and AST, but these methods may lack robustness when handling complex dependencies across multiple modules or files. Therefore, it is recommended that the author consider adding robustness validation for the fault localization results. Additionally, while this method may perform well for specific code structures and programming styles, its effectiveness remains to be tested in scenarios with significant code variation or cases requiring deep dynamic analysis.

**Questions:**

The main points and issues have been outlined in the "Weaknesses" section.

---

### Official Review · Reviewer_pq5J · 2024-11-03

**Soundness:** 2
**Presentation:** 2
**Contribution:** 2
**Rating:** 1
**Confidence:** 4

**Summary:**

The paper proposes an algorithm for using LLM-based code agents to repair programs given a task description (Github issue) and a candidate solution (pull request). The main contributions are algorithmic and empirical. The approach consists of using fault localization techniques to identify potentially buggy code blocks. These are then patched by an LLM using a debugging workflow in a loop. Experiments are performed on the SWE-bench dataset. Results show that the proposed algorithm is better than two baselines (SWE-agent) and AutoCodeRover at solving the code task and repairing the candidate program.

UPDATE: After reading the other reviews, it seems all reviewers have raised serious questions about the paper. There is no author response. I don't think the paper is ready for publication at this time and I've reduced my score further to indicate that.

**Strengths:**

- The paper tackles an important and interesting problem. Improving code agents is likely to have a large impact and be of significant interest to the community.

- The proposed approach is intuitively clear. The main ideas are easy to follow.

- There seems to be some algorithmic novelty to the approach (but I'm not certain).

- The results seem to suggest a significant performance improvement over two recent baselines.

**Weaknesses:**

- While the description of the proposed approach in Sec 2.2 and 2.2 sounds fine, there is insufficient technical detail about nearly each of the sub-components proposed in the method. The questions below contain more details but a quick summary includes the implementation of the fault localization (suspiciousness score, refining symbol indexing, code segment ranking), LLM implementation details (LLM, prompt) and implementation details of the debugging loop (budgets and timeouts, token and time costs, number of patches per example, etc.). Without these details, it becomes challenging to evaluate the paper for technical soundness and novelty. Please consider significantly expanding the amount of technical and implementation detail in the main paper and include appendixes with LLM details.

- The paper could do a better job of highlighting which components of the approach are novel, which build on prior work and which are reused from prior work. New code agents are rapidly being proposed in this very active area of research and the paper needs to do a better job of placing itself in this growing body of work.

- On first reading, I thought the performance improvement over the baselines might be coming from the use of the LLM in the debugging loop. However, the ablation study in Table 3 shows very little performance regression when the debugging phase (Sec 2.3) is removed. The issue is compounded on Line 338 stating "The ablation study highlights the importance of both the program call graphs and the simulated debugging mode in our method". This reader is left confused as to the exact sources of the performance improvement over the baselines. The issue is compounded by the lack of implementation detail, examples and traces, and error analysis.

- Overall, something interesting and useful might be happening in the paper but I can't be confident with this level of detail. Fixing this in the rebuttal period is likely to cause a very large delta and has a good chance of raising more questions. This leads me to recommend rejection in this round.

**Questions:**

1. (Line 177) How is the weight $w_{i,t}$ computed in the SBFL suspiciousness score?

2. In Section 2.1.4, what is the exact implementation of each of these?
   - refining symbol indexing
   - referencing analysis techniques
   - mapping the problem description to code segments
   - use of dependency relationships
   - ranking of code segments

3. (Line 225) What is `R`? How is it used in `UpdateTests`?

4. (Line 328) In Table 3, why is the result of the ablation study not showing significant performance drop (relative to the baselines) when call graphs and simulated debugging is excluded? Where is all the performance coming from (compared to prior work)? Please discuss in more detail.

5. Why does Table 1 and Table 3 report results on SWE-bench lite when Table 2 uses SWE-bench? Is there a reason to not just use the full set of instances in SWE-bench?

6. How exactly is a LLM used in Algorithm 1? What are the LLM implementation details? Choice of LLM, prompts, etc.

7. What are the computational considerations of the debugging loop? Time and token budgets? Retries?

---

### Note · Authors · 2024-11-28

I have read and agree with the venue's withdrawal policy on behalf of myself and my co-authors.